

# Integrated platform for structural and functional analysis of terpene synthases of *Cannabis sativa*

Danielle Wiles[1,2], James Roest[3], Bhuvana Shanbhag[1,2], Julian Vivian[3,4,5] and Travis Beddoe[1,2]

[1] Department of Ecological, Plant and Animal Science, La Trobe University, Bundoora, Victoria, Australia
[2] Australian Research Council Research Hub for Medicinal Agriculture, La Trobe University, Bundoora, Victoria, Australia
[3] St Vincent's Institute of Medical Research, Fitzroy, Victoria, Australia
[4] Australian Catholic University, Fitzroy, Victoria, Australia
[5] Department of Medicine, University of Melbourne, Melbourne, Victoria, Australia

Corresponding author
Travis Beddoe,
t.beddoe@latrobe.edu.au

## ABSTRACT

Terpenoids are the largest and most diverse family of natural products. Volatile terpenes from *Cannabis sativa* are crucial in flavours, fragrances, and pharmaceuticals due to their unique odours and biological activities, including antimalarial, antibacterial, and insecticidal properties. Their synthesis is catalysed by terpene synthase (TPS) enzymes, which perform cyclisation and rearrangement reactions of over 55,000 distinct terpene compounds. However, low catalytic efficiency of *C. sativa* TPSs limits their use in large-scale commercial production. The complex biochemistry of these reactions is not well understood due to limited enzyme structure information. To address this, we have developed an integrated platform for the systematic expression, purification, enzymatic characterisation, and crystallisation of TPS enzymes from *C. sativa*. This workflow combines kinetic, thermostability, and structural analyses with a data-mining-informed crystallisation screen that enabled the production of diffraction-quality crystals. As a demonstration of the platform's utility, ten *C. sativa* TPS enzymes were functionally characterised, revealing turnover rates ($k_{cat}$) ranging from 0.0011 to 0.0204 s$^{-1}$ and diverse substrate specificities, with each enzyme producing a distinct product profile, highlighting the need for systematic characterisation of *C. sativa* terpene biosynthesis. Our findings provide a framework for the structural and functional study of *C. sativa* TPSs. The developed platform sets the stage for future metabolic engineering aimed at optimising terpene production for pharmaceutical, pest management, and synthetic biology applications.

## INTRODUCTION

*Cannabis sativa* (cannabis) has been utilised for thousands of years as a source of fibre, food, and oil, as well as a medicinal agent and a recreational intoxicant (*Bonini et al., 2018*; *Hui-Lin, 1974*; *Kalant, 2001*; *Zuardi, 2006*). Today, it is highly valued for its

pharmacologically active specialised metabolites, including cannabinoids, monoterpenes, and sesquiterpenes. These bioactive compounds are predominantly concentrated in the resin of glandular trichomes found in female cannabis inflorescences (*Andre, Hausman & Guerriero, 2016*). While research has focused on the bioactivity of cannabinoids, the terpenes are increasingly being examined for their bioactive properties and commercial value. Cannabis is a prolific producer of terpenoids, with over 230 distinct compounds identified across various tissues (*Aizpurua-Olaizola et al., 2016*; *Downer, 2020*; *Hanuš & Hod, 2020*; *Roell, 2020*).

Terpenes constitute the largest class of plant specialised metabolites, playing a crucial role in plants' aroma and flavour profiles (*Jörg, Meyer-Gauen & Croteau, 2017*; *Karunanithi & Zerbe, 2019*; *Theis & Lerdau, 2003*). Terpenoids have a variety of commercial applications as therapeutics, cosmetics, flavouring agents, fragrances, agrochemicals, and disinfectants (*Ajikumar et al., 2008*; *Cox-Georgian et al., 2019*; *Masyita et al., 2022*; *Nuutinen, 2018*; *Paduch et al., 2007*). In cannabis, the plant's characteristic scent and flavour are derived from the unique combination of structurally diverse terpenoids, influencing consumer preferences, particularly among recreational and medicinal cannabis users (*Oswald et al., 2023*). Additionally, there is growing evidence that terpenes elicit various medicinal properties that affect both humans and animals (*Russo, 2011*), prompting efforts to breed cannabis cultivars with specific terpene profiles (*Barcaccia et al., 2020*; *Chandra, Lata & ElSohly, 2017*; *Grof, 2018*; *Rocha et al., 2020*).

Terpenes are synthesised from the 5-carbon precursors, isopentenyl diphosphate (IPP) and dimethylallyl diphosphate (DMAPP), which are produced *via* the mevalonate (MVA) pathway in the cytosol or the methylerythritol phosphate (MEP) pathway in plastids. These precursors are further condensed to form prenyl diphosphate intermediates, such as geranyl diphosphate (GPP) and farnesyl diphosphate (FPP), which are then converted into monoterpenes ($C_{10}$) and sesquiterpenes ($C_{15}$), respectively by terpene synthase (TPS) enzymes (*Christianson, 2017*; *Lange et al., 2000*). The remarkable structural and chemical diversity exhibited by terpenes is due to the vast array of reactions catalysed by TPSs, which can generate multiple products from a single precursor through cyclisation, rearrangement, and hydride shifts (*Karunanithi & Zerbe, 2019*). This diversity is further enhanced by the enzyme's active site architecture, which stabilises different carbocation intermediates, making TPSs essential for the biosynthesis of the complex and varied terpene landscape observed in nature (*Christianson, 2017*).

TPS enzymes are encoded by large gene families, typically comprising 20–150 genes in most plant species, enabling them to exhibit remarkable variability in both substrate preference and product specificity (*Tholl, 2006*; *Zhou & Pichersky, 2020*). This unique versatility allows them to generate multiple terpene products from a single substrate or utilise various substrates to produce a variety of different terpenes (*Li & Tao, 2024*; *Kampranis et al., 2007*). Such evolutionary plasticity means that even single amino acid changes within the active site can significantly alter the enzyme's product profile (*Bohlmann & Gershenzon, 2009*; *Schilmiller et al., 2009*; *Zhou & Pichersky, 2020*). However, given the complex evolutionary history of TPS, the product profile of a given enzyme cannot reliably be predicted based on sequence similarity alone and, therefore, requires

functional characterisation of individual TPS enzymes to determine the specificities of their catalysis.

In cannabis, at least 55 TPS genes have been identified in the genome, reflecting the extensive diversity within this gene family (*Allen et al., 2019*; *Booth et al., 2020*; *Booth, Page & Bohlmann, 2017*; *Xu et al., 2024*). The overall gene structure of individual TPS genes is remarkably well conserved across cannabis TPS subfamilies; however, gene length varies (*Allen et al., 2019*; *Xu et al., 2024*). Despite the functional characterisation of several cannabis TPS genes (*Booth et al., 2020*; *Booth, Page & Bohlmann, 2017*; *Günnewich et al., 2007*; *Livingston et al., 2020*; *Zager et al., 2019*), the inability to precisely link specific TPS enzymes to their terpene products poses a major challenge. This limitation hinders efforts to generate targeted or novel terpene profiles and to optimise their production for industrial applications. Furthermore, the lack of comprehensive kinetic, thermostability, and structural analyses prevents a deeper understanding of TPS functionality and restricts their potential for metabolic engineering.

Engineering TPSs for targeted terpene production presents a transformative solution to the challenges of traditional terpene extraction methods, which are constrained by strict regulatory frameworks, inconsistent metabolite profiles due to environmental variability, and the inherent complexity of cannabis-specialised metabolites. To overcome these challenges, the present study was designed with the following objectives: (1) to establish a robust, reproducible platform for the heterologous expression and purification of *C. sativa* TPS enzymes; (2) to perform systematic kinetic and thermostability characterisation of these enzymes to evaluate their catalytic efficiency and functional diversity; (3) to develop a directed crystallisation screening method informed by data mining of known conditions to obtain high-quality TPS crystals; and (4) to apply this integrated approach to a representative set of ten *C. sativa* TPSs in order to elucidate structure–function relationships and identify determinants of substrate specificity. This research deepens our understanding of terpene biosynthesis and lays the groundwork for future engineering of TPS enzymes to enhance their catalytic efficiency and product specificity, thereby paving the way for optimised terpene production at industrial scales.

# MATERIALS AND METHODS

## Cloning of CsTPS

Expression constructs encoding three full-length sesqui-TPS (CsTPS9FN, CsTPS16CC and CsTPS20CT), two mono/sesqui-TPS (CsTPS5FN and CsTPS19BL), and five mono-TPS sequences (CsTPS12PK, CsTPS13PK, CsTPS3FN, CsTPS1SK and CsTPS37FN), which had their plastidial-targeting sequence motif truncated (Table 1), were synthesised (Twist Bioscience) as codon-harmonised (Codon Wizard) and codon-optimised for *Escherichia coli*. The genes were cloned into a pET28a$^+$ expression vector (*Luna-Vargas et al., 2011*), using *NdeI* and *XhoI* restriction sites.

## Protein expression and purification

Recombinant TPS enzymes were expressed in *E. coli* BL21(DE3) *T7 Express lysY/Iq* competent cells (New England Biolabs, Ipswich, MA, USA) using the NEB high-efficiency

**Table 1 Summary of CsTPS enzymes used in study.**

| Functional gene ID* | GenBank ID | Proposed functionality | *C. sativa* cultivar | TPS type | DNA seq. length (bp) | Protein seq. length (AA) | Predicted MW (kDa) | Theoretical pI |
|---|---|---|---|---|---|---|---|---|
| CsTPS3FN | KY014561 | β-myrcene synthase | Finola | Mono | 1,692* | 584 | 68.90 | 6.02 |
| CsTPS1SK | ABI21837 | (-)-limonene synthase | Skunk | Mono | 1,641* | 567 | 66.32 | 6.11 |
| CsTPS37FN | KY014554 | terpinolene synthase | Finola | Mono | 1,692* | 584 | 68.68 | 5.77 |
| CsTPS12PK/ CsTPS33PK | KY624371 | α-terpinene, γ-terpinene synthase | Purple Kush | Mono | 1,854 | 633 | 73.75 | 5.46 |
| CsTPS13PK | KY014558 | (Z)-β-ocimene synthase | Purple Kush | Mono | 1,803 | 616 | 72.27 | 6.11 |
| CsTPS9FN | KY014555 | β-caryophyllene/α-humulene synthase | Finola | Sesqui | 1,704 | 587 | 68.88 | 6.00 |
| CsTPS16CC | MK131289 | germacrene-B synthase | Cherry Chem | Sesqui | 1,716 | 591 | 69.55 | 6.43 |
| CsTPS20CT | MK801762 | hedycaryol synthase | Canna Tsu | Sesqui | 1,656 | 571 | 66.84 | 6.29 |
| CsTPS5FN | KY014560 | β-myrcene/(-)-α-pinene synthase | Finola | Mono/Sesqui | 1,722 | 593 | 69.43 | 6.04 |
| CsTPS19BL | MK801763 | nerolidol/linalool synthase | Black Lime | Mono/Sesqui | 1,665 | 574 | 66.55 | 6.30 |

Note:
* Plastidial targeting sequence truncated.

transformation protocol. Briefly, 25 μl aliquots of competent cells were mixed with 2 μl of plasmid DNA by heat shock and incubated in Luria-Bertani (LB) medium (1.0% (w/v) tryptone, 0.5% (w/v) yeast extract, 1.0% (w/v) NaCl) at 37 °C for 1 h with shaking at 200 rpm. Transformed cells were plated on LB-agar containing 50 μg ml$^{-1}$ kanamycin and incubated overnight at 37 °C.

A single colony was used to inoculate a 10 mL starter culture in Terrific Broth (TB; 1.2% (w/v) tryptone, 2.4% (w/v) yeast extract, 0.4% (v/v) glycerol, 17 mM KH$_2$PO$_4$, 72 mM K$_2$HPO$_4$) with 50 μg ml$^{-1}$ kanamycin and grown overnight at 37 °C with shaking at 200 rpm. The starter culture (1:50 dilution) was used to inoculate 400 mL TB medium supplemented with kanamycin (50 μg ml$^{-1}$). Cultures were grown at 37 °C for 16 h, induced with 1 mM isopropyl β-D-1-thiogalactopyranoside (IPTG), and incubated at 16 °C for an additional 16 h. Cells were harvested by centrifugation at 10,330 × g for 30 min at 4 °C.

Cell pellets were resuspended in 30 mL lysis buffer (20 mM Tris-HCl (pH 7.5), 500 mM NaCl, 10 mM imidazole) and lysed by sonication (Ultrasonics, Brookfield, CT, USA) at 40% amplitude for 30-s bursts with 30-s rest intervals repeated three times, on ice. Lysates were clarified by centrifugation at 10,330 x *g* at 4 °C for 20 min. The process was repeated to ensure maximal removal of cell debris. The clarified lysate was syringe-filtered (0.22 μm), prior to purification.

Cleared lysates were loaded onto a gravity column containing 2 mL of 50% (v/v) of Ni-NTA resin slurry (Thermo Fisher Scientific, Waltham, MA, USA), pre-equilibrated with wash buffer (20 mM Tris-HCl (pH 7.5), 500 mM NaCl, 10 mM imidazole). The column was washed twice with 30 mL wash buffer, and TPS proteins were eluted with 5 mL elution buffer (50 mM Tris-HCl (pH 7.5), 500 mM NaCl and 500 mM imidazole) in 1 mL

fractions. Each step of the purification process was analysed by sodium dodecyl sulphate–polyacrylamide gel electrophoresis (SDS–PAGE) and Coomassie blue staining.

Ni-affinity eluted fractions containing the proteins of interest were pooled and further purified by size-exclusion chromatography (SEC) using a HiPrep 16/60 Superdex 200 column (GE Healthcare Life Sciences, Marlborough, MA, USA) on a ÄKTA Basic Fast Protein Liquid Chromatography (FPLC) system. The gel filtration buffer comprised 25 mM 4-(2-Hydroxyethyl) piperazine-1-ethane-sulfonic acid (HEPES; pH 7.0), 10 mM $MgCl_2$, 100 mM KCl and 1 mM Dithiothreitol (DTT) and all steps were performed at a flow rate of 1 mL $min^{-1}$. Peak fractions containing the CsTPS enzymes (>95% pure as judged by SDS–PAGE) were combined and concentrated to 10 mg $mL^{-1}$ using Amicon centrifugal filters (10 kDa molecular weight cut off, Millipore). Protein concentrations were measured using Bradford's method (*Bradford, 1976*) and stored at 4 °C.

## Enzyme activity assay

Enzymatic activity assays were performed as described by *Allen et al. (2019)* with minor modifications. TPS activities were assayed in triplicate in a final volume of 500 μL assay buffer (20 mM HEPES (pH 7.5), 100 mM KCl, 5 mM $MgCl_2$, 1 mM dithiothreitol (DTT), 10% (v/v) glycerol), 100 μM geranyl diphosphate (GPP), geranyl-geranyl diphosphate (GGPP) or farnesyl diphosphate (FPP) (Merck, Sigma Inc., St. Louis, MO, USA) and purified protein. 500 μL of hexane (Sigma-Aldrich) containing 2.5 μM isobutylbenzene as an internal standard was overlaid to trap the volatile products. Reactions were incubated at 30 °C for 16 h and vortexed for 30 s. Volatile products were extracted by centrifugation at 1,000 x *g* for 30 min at 4 °C. Boiled enzyme controls were included to determine the background noise of the assay.

## Product identification by gas-chromatography mass spectrometry

Gas-chromatography mass spectrometry (GC-MS) analysis was conducted using a Thermo Scientific system equipped with a Trace 1310 GC interfaced with a triple quadrupole MS TSQ 8000 Evo and a TriPlus Robotic Sample Handling autosampler (Thermo Fisher Scientific, Waltham, MA, USA). The column was a Thermo Scientific TG-5SILMS capillary column (60 m × 0.25 mm ID, 1.0 μm film thickness), and carrier gas was He at a constant flow rate of 1.5 mL $min^{-1}$. The inlet temperature was 280 °C with a split ratio of 5:1, and the injection volume was 1 μL. The initial oven temperature was set at 50 °C with 5 min hold time, then increased to 300 °C at a rate of 5 °C $min^{-1}$, and held at 300 °C for 5 min. The MS was set in full scan mode with a mass range of 35–400 amu, delay time of 10.5 min, and ionisation by electron impact with ionisation energy of 70 eV. The ion source temperature was 230 °C, and the MS transfer line temperature was 280 °C. Gerstel Maestro software v1.5 controlled the autosampler, and data were acquired using Thermo Scientific XCalibur Software v4.7. Data was processed using Thermo Scientific XCalibur Qual Browser v4.7 or FreeStyle v1.8 (Thermo Fisher Scientific, Waltham, MA, USA).

To confirm the presence of most terpenes, authentic standards (Sigma, St. Louis, MO, USA) were used during the analysis. However, the sesquiterpenes δ-elemene, β-elemene,
γ-elemene, epi-β-caryophyllene, alloaromadendrene, elemol, germacrene-D, guaiol, globulol, γ-eudesmol, and α-eudesmol were identified tentatively through comparisons with the NIST library due to the lack of available internal standards (Fig. S4). Similarly, several monoterpenes, including β-phellandrene, allo-ocimene, fenchol, β-terpineol, pinene hydrate, and geranyl methyl ether, were also putatively identified *via* NIST library comparisons. While geranyl methyl ether and pinene hydrate were detected as chromatographic peaks and labelled accordingly, these compounds are neither monoterpenes nor sesquiterpenes—the primary focus of this study—and were therefore excluded from the final quantification percentages.

## Malachite green assay for kinetic measurements

Kinetic parameters were determined using the malachite green assay (*Vardakou et al., 2014*). Reactions were performed in triplicate in 96-well flat-bottomed plates (Greiner Bio-One, Kremsmünster, Austria). Standard curves were established for both monophosphate (Pi) and pyrophosphate (PPi) using serial 2-fold dilutions ranging from 0.01 to 50 μM. For *kcat* determination, reactions (50 μL) contained malachite green assay buffer (25 mM 2-(N-morpholino) ethanesulfonic acid (MES), 25 mM 3-(Cyclohexylamino)-1-propanesulfonic acid (CAPS), 50 mM Tris, 5 mM MgCl2, pH 7.5), 25 mU of inorganic pyrop/hosphatase from *Saccharomyces cerevisiae* (Sigma, St. Louis, MO, USA), 100 μM substrate (FPP or GPP), and serial 2-fold dilutions of protein (0.003–0.2 μM). Reactions were incubated at 37 °C for 30 min prior to being terminated by the addition of the malachite green development solution (prepared according to *Pegan et al. (2009)*) and further incubated for 15 min before measuring absorbance at 623 nm using a plate reader (Bio-Rad, Hercules, CA, USA). Steady-state kinetic measurements were also conducted in triplicate in a 50 μL reaction volume containing malachite green assay buffer, a fixed concentration of 0.014 μM of the protein of interest, and serial 2-fold dilutions of the pyrophosphate substrate (FPP or GPP) ranging from 0.02–100 μM. Reactions were also incubated at 37 °C for 30 min before termination and absorbance measurement at 623 nm, following the same procedure described above The kinetic parameters ($V_{max}$, $K_M$ and $k_{cat}$) were obtained from non-linear regression analysis of the data using the Michaelis–Menten model in GraphPad Prism.

## Thermofluor protein stability assay

For downstream applications, purified CsTPS enzymes were concentrated to 10 mg ml$^{-1}$. To optimise storage conditions an 80-condition buffer screen (Table S1) was developed, based on commonly used buffers for TPS enzymes. The temperature at which CsTPS unfolding occurred for each of the conditions ($T_m$ values) was compared with the original storage buffer, and changes in unfolding temperature ($\Delta T_m$) were calculated. Thermal denaturation was monitored using SYPRO Orange dye (Sigma) detected on a real-time PCR machine. Melting temperatures ($T_m$) were calculated as described previously (*Ericsson et al., 2006*). Briefly, solutions of 7.5 μL of 300 x SYPRO Orange (Merck Sigma, St. Louis, MO, USA), 12.5 μL of test buffer conditions (Table S1), and 5 μL of 2.5 mg mL$^{-1}$ CsTPS proteins were added to the wells of a 384-well thin wall PCR plate (Bio-Rad,

Hercules, CA, USA). Water was added instead of a buffer in the control samples. The plates were sealed with Optical-Quality Sealing Tape (Bio-Rad, Hercules, CA, USA) and heated in an ABI QuantStudio 5Dx Real-Time qPCR System from 20 to 95°C in increments of 1 °C. Fluorescence changes in the wells of the late were monitored simultaneously with a charge-coupled device (CCD) camera. The wavelengths for excitation and emission were 490 and 575 nm, respectively.

### Creation of the TPS-crystallisation screen

A 48-condition screen specifically directed towards the crystallisation of TPSs was developed. To create this screen, the Protein Data Bank (PDB; *Berman et al., 2000*), was searched to identify the crystallisation conditions of TPSs and related proteins. Crystallisation conditions for 120 deposited structures of proteins were returned. Individual searching of each entry led to the removal of 16 entries that had not published crystallisation conditions, leaving the final count of crystallisation conditions at 104. Various conditions from these entries were analysed, such as pH, buffer, precipitant, and salt to develop a TPS-specific sparse matrix screen containing 48 different conditions (Table S2).

### Crystallisation and optimisation of TPS crystals

After creating the TPS-crystallisation screen, crystallisation experiments were performed. Each of the 10 CsTPS proteins were screened through the 48 conditions of the newly created TPS-crystallisation screen. Crystallisation was performed using the hanging-drop vapour diffusion method at 293 K. Briefly, a 1 µL drop of protein solution (10 mg mL$^{-1}$ CsTPS) was mixed with a 1 µL drop of precipitant solution (Table S2) and equilibrated against a 500 µL reservoir of the precipitant solution.

## RESULTS

### Development of a pipeline for the characterisation of CsTPS enzymes

Existing protocols for TPSs overexpression in bacterial systems often suffer from challenges such as protein misfolding, formation of inclusion bodies, low yield and stability issues (*Raman et al., 2014*; *Wiles et al., 2022*). To overcome these limitations, we established a robust pipeline for the overexpression, functional and structural characterisation of TPS's (Fig. 1). Currently, 55 different TPS have been identified in the Cannabis genome (*Allen et al., 2019*). We selected 10 CsTPS to act as candidates in our structural and functional pipeline (Table 1). The selection criteria included representing different TPS types (mono-TPS and sesqui-TPS), including TPSs that can utilise multiple substrates or produce multiple products, and encompassing enzymes from both drug and fibre-type cannabis from a broad selection of different cultivars.

### Recombinant expression and purification of CsTPS enzymes

The ten candidate CsTPS enzymes were successfully expressed recombinantly in *E. coli* and purified using immobilised metal-ion affinity chromatography (IMAC). SDS-PAGE analysis confirmed the presence of a single protein band with the expected molecular mass

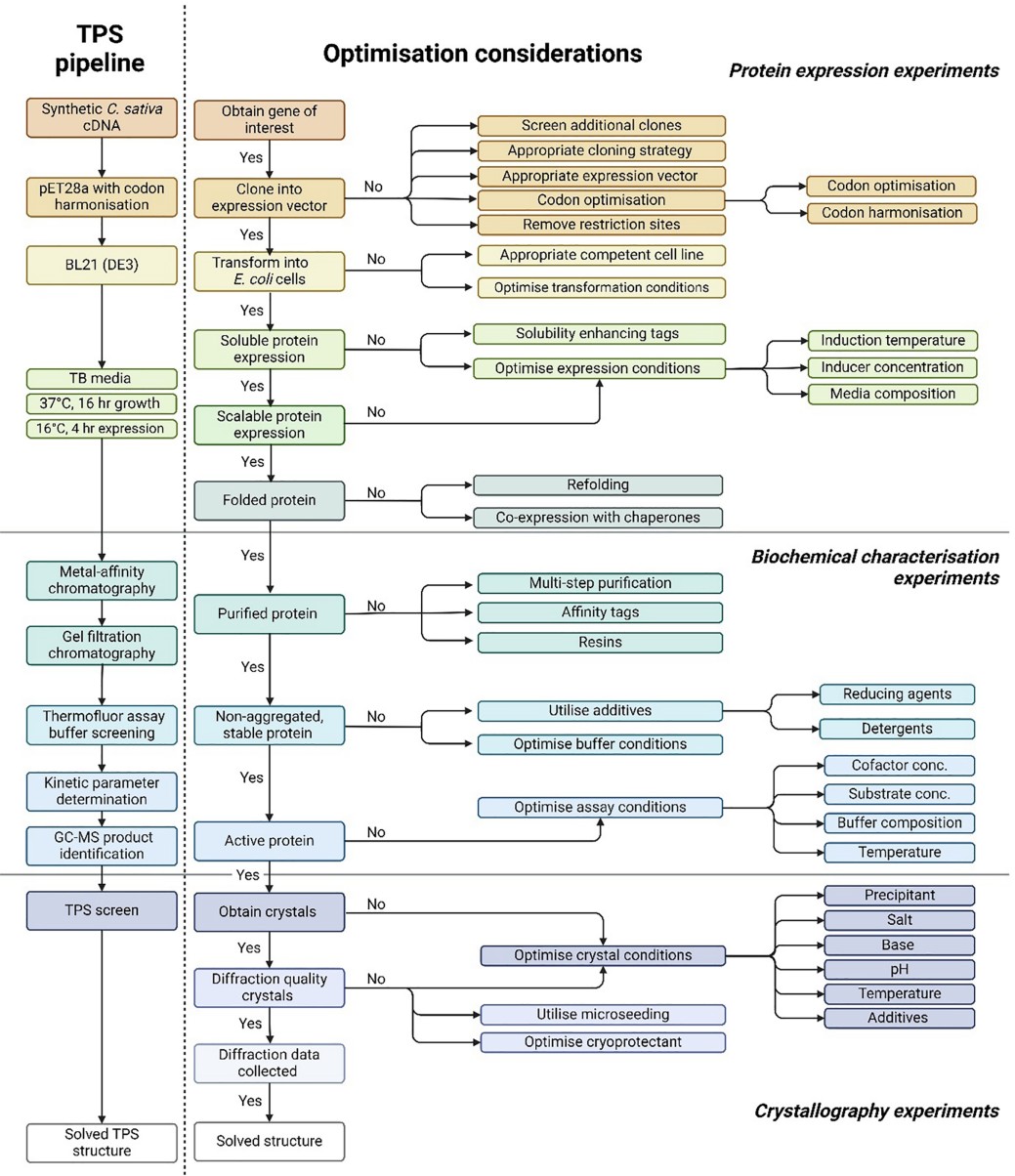

**Figure 1 Pipeline for the functional and structural characterisation of *Cannabis sativa* terpene synthase enzymes.** The comprehensive methodology for terpene synthase crystallography, highlighting the critical pre-crystallisation screening steps essential for successful crystal formation. The approach integrates key information from protein expression, biochemical characterisation, and secondary structure prediction, facilitating the design of optimised protein constructs and crystallisation conditions to enhance the likelihood of obtaining high-quality terpene synthase crystals.

of 66.32–73.75 kDa for each enzyme, indicating that all ten CsTPS proteins were expressed and purified in their soluble forms (Fig. S1). After initial ion-affinity purification, yields of each CsTPS ranged from 2 to 25 mg L$^{-1}$. To further purify and characterise all CsTPS were subjected to size-exclusion chromatography (SEC). A typical trace chromatogram

for each of the CsTPS (Fig. S2) revealed a single peak corresponding to a molecular weight of approximately 66.32–73.75 kDa, suggesting that all CsTPS are monomers in solution.

## Impact of optimal buffer conditions for CsTPS stability

The purified CsTPS enzymes were concentrated for structural studies. However, different levels of protein precipitation occurred after overnight storage. To overcome this problem, a thermofluor assay was used to determine the optimal storage buffer. Relatively large variations in stability could be observed when varying the buffers (Fig. 2), and a few buffers appear to be generally more favourable for protein stabilisation (Fig. 2B; Table 2). Tris pH 8.0 and 8.5, and HEPES pH 7.5 were the most stabilising buffers, whereas MES pH 5.5, Bis-Tris pH 6.0 and CAPS pH 9.0 significantly destabilised several of the CsTPS proteins (Table 2, Fig. 2B). Several of the 56 buffer conditions investigated in the buffer screen gave no measurable transitions in combination with one or a few of the proteins, possibly caused by destabilisation or partial unfolding and potential aggregation of the proteins. For example, a clear thermal transition could be detected with only 6 of the 10 proteins in combination with CAPS pH 9.0. However, for some of the buffers, among them the overall most stabilising buffers, a measurable transition could be recorded together with all the proteins. The three most stabilising buffers were determined for each CsTPS protein, and Tris pH 8.0 was consistently found to improve the stability of the CsTPS enzymes. This contrasts with both the literature, where buffers with a pH of 7–7.5 are generally used for TPS proteins. While a pH of 8.0 is above the theoretical isoelectric point (PI), of approximately 5–6, which generally helps maintain solubility by increasing the net negative charge on the protein, it is notable that a slightly more alkaline conditions provided greater stability than the commonly used neutral to mildly acidic buffers. Given that the median $\Delta$Tm for Tris pH 8.0 with the addition of the salt additives 100 and 200 mM NaCl were found to be more than 4 °C, these buffer conditions were considered to improve the stability of the CsTPS proteins significantly and were deemed suitable buffers for continuing functional characterisation of the enzymes.

## Product profile analysis of CsTPS

To determine if the recombinant CsTPS were enzymatically active, all recombinant CsTPS were assayed as purified proteins with GGPP, GPP and FPP, and the reaction products were analysed by GC-MS. No products were seen for any CsTPS when assayed with GGPP, indicating that none of the CsTPS enzymes analysed form diterpenes.

### Monoterpene synthases (GPP assay)

Five of the ten chosen CsTPS had previously been identified as proposed monoTPSs, while two were suggested to have dual activity with both GPP and FPP as substrates (Table 1). The five monoTPSs (CsTPS1SK, CsTPS3FN, CsTPS12PK, CsTPS13PK, and CsTPS37FN) exhibited enzymatic activity, producing a diverse array of monoterpenes (Fig. 3, Table 3). CsTPS1SK predominantly produced limonene (74.72%), in line with previous studies (*Günnewich et al., 2007*), but we also detected the minor products β-pinene (5.18%), α-terpineol (4.83%), and β-terpineol (2.30%) (Fig. 3; Table 3). Interestingly, we also

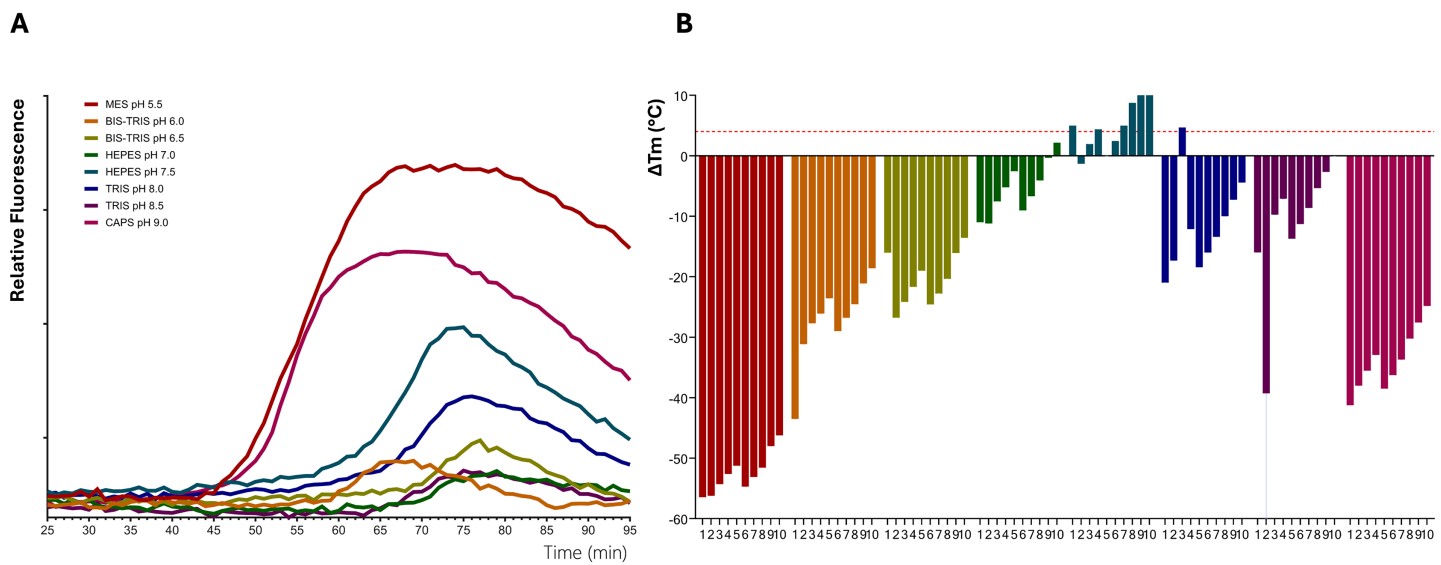

**Figure 2 Thermal shift assay for improving protein stability of terpene synthases from *Cannabis sativa*.** (A) Representative thermofluor melt curve for terpinolene synthases from *C. sativa* (CsTPS37FN). (B) Changes in the unfolding transition temperature (ΔTm) were calculated for each CsTPS protein in 80 buffer conditions. The bars represent the median ΔTm values. 1. No addition of salt; 2. 50; 3. 100; 4. 200; 5. 50; 6. 100; 7. 200; 8. 50; 9. 100; 10. 200 mM MgCl. A negative ΔTm value signifies that the buffer destabilises the proteins, and a positive ΔTm value indicates that the buffer has a stabilising effect. ΔTm values more than 4 °C are considered significantly stabilising, as indicated by the horizontal dashed red line.

**Table 2 Summary of most stabilising buffer conditions resulting from the thermofluor melt curve for each of the *Cannabis sativa* terpene synthase (CsTPS) candidates.**

| TPS Name | Best buffer condition | $T_m$ | $\Delta T_m$ |
|---|---|---|---|
| CsTPS3FN | 0.2 M Tris pH 8.0 0.1 M NaCl | 81.13 °C | 4.67 °C |
| CsTPS1SK | 0.2 M HEPES pH 7.5 0.1 M KCl | 80.2 °C | 3.74 °C |
| CsTPS5FN | 0.2 M HEPES pH 7.0 0.1 M KCl | 81.89 °C | 5.43 °C |
| CsTPS37FN | 0.2 M Tris pH 8.0 0.1 M NaCl | 81.25 °C | 4.79 °C |
| CsTPS9FN | 0.2 M HEPES pH 7.5 0.1 M KCl | 83.18 °C | 6.72 °C |
| CsTPS16CC | 0.2 M HEPES pH 7.5 0.1 M KCl | 79.36 °C | 2.9 °C |
| CsTPS20CT | 0.2 M HEPES pH 7.0 0.1 M NaCl | 80.2 °C | 3.74 °C |
| CsTPS19BL | 0.2 M Tris pH 8.0 0.1 M NaCl | 80.57 °C | 4.11 °C |
| CsTPS12PK | 0.2 M Tris pH 8.0 0.1 M KCl | 81.03 °C | 4.57 °C |
| CsTPS13PK | 0.2 M HEPES pH 7.5 0.1 M KCl | 81.67 °C | 5.21 °C |

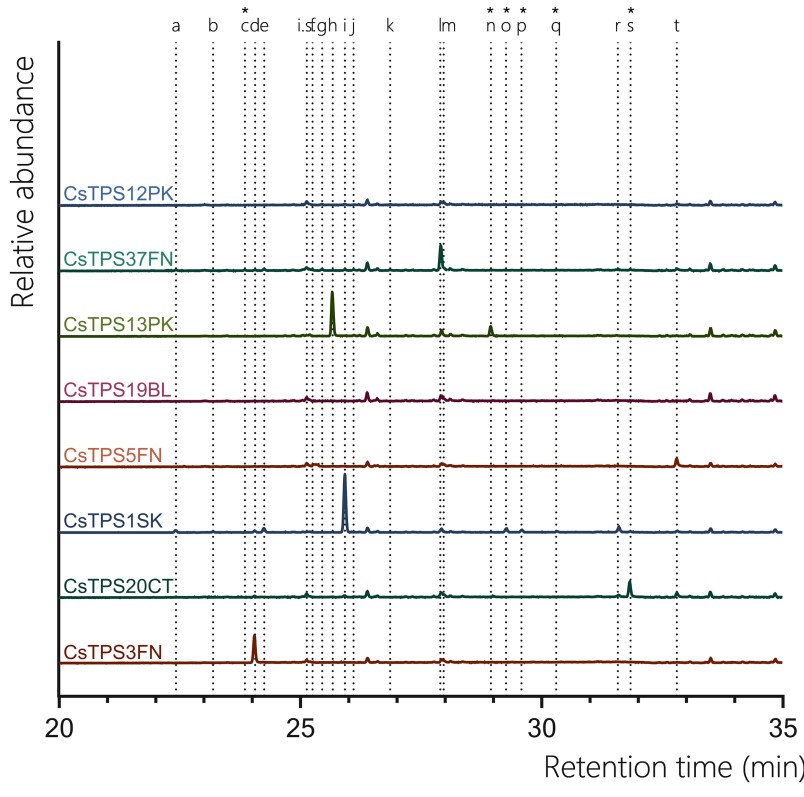

**Figure 3 GC-MS traces showing the monoterpene products of CsTPS.** Traces show GC-MS total ion chromatogram from CsTPS assays with GPP. CsTPS3FN: β-myrcene synthase; CsTPS16CC: Germacrene B synthase; CsTPS1SK: Limonene synthase; CsTPS5FN: Myrcene/pinene synthase; CsTPS19BL: Nerolidol/Linalool synthase; CsTPS13PK: Ocimene synthase; CsTPS37FN: Terpinolene synthase; CsTPS12PK: Terpinene synthase. Peaks: (a) α-pinene, (b) camphene (c) β-phellandrene* (d) β-myrcene, (e) β-pinene, (f) d-carene, (g) α-terpinene (h) (E)-β-ocimene (i) limonene, (j) (Z)-β-ocimene, (k) g-terpinene, (l) terpinolene, (m) linalool, (n) allo-ocimene*, (o) fenchol*, (p) β-terpineol*, (q) pinene hydrate*, (r) α-terpineol, (s) geranyl methyl ether*, (t) geraniol, i.s. = internal standard. *No reference standard available, putative identification of compound using National Institute of Standards and Technology (NIST) library.

identified camphene (0.98%) and fenchol (3.55%), which were not previously reported for this enzyme (Fig. 3, Table 3). CsTPS3FN exhibited strict specificity for β-myrcene, producing this monoterpene at 100% when incubated with GPP, consistent with previous reports (Fig. 3, Table 3; *Booth, Page & Bohlmann, 2017*). CsTPS5FN also produced myrcene as its most abundant monoterpene product (37.09%) (Fig. 3, Table 3). In addition to β-myrcene, several other monoterpenes were produced, including (-)-α-pinene (23.45%), (-)-limonene (17.33%), sabinene (15.26%), and (-)-β-pinene (8.87%), consistent with previous reports (Fig. 3, Table 3; *Booth, Page & Bohlmann, 2017*). CsTPS12PK predominantly produced α-terpinene (68.42%), a result consistent with previous studies (*Booth, Page & Bohlmann, 2017*), while also generating minor amounts of γ-terpinene (15.29%) and α-phellandrene (4.21%) (Fig. 3, Table 3). Interestingly, our study also identified trace amounts of sabinene (3.87%), which had not been previously reported for this enzyme (Fig. 3, Table 3). CsTPS13PK was highly specific for (Z)-β-ocimene,

**Table 3 Terpene products formed by recombinant *Cannabis sativa* terpene synthase (CsTPS) enzymes.**

| TPS enzyme | Substrate | Terpenes produced | Percent total (%) |
|---|---|---|---|
| **Monoterpene synthases** | | | |
| CsTPS3FN | GPP | β-myrcene | 100.00 |
| CsTPS1SK | GPP | α-pinene | 2.98 |
| | | camphene | 0.98 |
| | | β-myrcene | 2.25 |
| | | β-pinene | 5.18 |
| | | limonene | 74.72 |
| | | terpinolene | 1.53 |
| | | fenchol* | 3.55 |
| | | β-terpineol* | 2.30 |
| | | α-terpineol | 4.83 |
| | | geraniol | 1.69 |
| CsTPS12PK | GPP | α-terpinene | 29.50 |
| | | limonene | 33.36 |
| | | γ-terpinene | 24.89 |
| | | β-myrcene | 12.25 |
| CsTPS13PK | GPP | (E)-β-ocimene | 79.52 |
| | | allo-ocimene | 1.80 |
| | | (Z)-β-ocimene | 18.68 |
| CsTPS37FN | GPP | α-pinene | 3.03 |
| | | β-phellandrene* | 2.17 |
| | | β-myrcene | 2.16 |
| | | β-pinene | 4.77 |
| | | delta-3-Carene | 3.67 |
| | | α-terpinene | 2.86 |
| | | limonene | 1.96 |
| | | y-terpinene | 1.13 |
| | | terpinolene | 70.69 |
| | | linalool | 2.95 |
| | | geraniol | 4.61 |
| **Sesquiterpene synthases** | | | |
| CsTP9FN | FPP | β-caryophyllene | 2.76 |
| | | humulene | 4.49 |
| | | epi-β-caryophyllene* | 67.30 |
| | | germacrene D* | 15.26 |
| | | globulol* | 10.19 |
| CsTPS16CC | FPP | β-elemene* | 1.05 |
| | | γ-elemene* | 3.82 |
| | | germacrene B* | 92.73 |
| | | alloaromadendrene* | 1.06 |

| Table 3 (continued) | | | |
|---|---|---|---|
| TPS enzyme | Substrate | Terpenes produced | Percent total (%) |
| **Mono/Sesquiterpene synthases** | | | |
| CsTPS19BL | GPP | linalool | 100.00 |
| | FPP | nerolidol | 100.00 |
| CsTPS5FN | GPP | α-pinene | 23.00 |
| | | β-myrcene | 37.00 |
| | | β-pinene | 8.00 |
| | | limonene | 17.00 |
| | | sabinene* | 15.00 |
| | FPP | farnesol* | 100.00 |
| CsTPS20CT | GPP | β-myrcene | 7.89 |
| | | limonene | 12.72 |
| | | (Z)-β-ocimene | 2.64 |
| | | terpinolene | 7.74 |
| | | α-terpineol | 26.06 |
| | | geraniol | 42.95 |
| | FPP | hedycaryol (elemol*) | 31.42 |
| | | guaiol | 19.96 |
| | | γ-eudesmol* | 20.51 |
| | | α-eudesmol* | 28.11 |

Note:
* No reference standard available, putative identification of compound using National Institute of Standards and Technology (NIST) library.

producing 94.11% of this product, which aligns with prior findings for this enzyme (Fig. 3, Table 3; (*Booth, Page & Bohlmann, 2017*)). Small amounts of α-ocimene (3.49%) were also detected, though this was not previously noted (Fig. 3, Table 3). For CsTPS37, the dominant product was terpinolene (81.62%), in agreement with past studies (*Livingston et al., 2020*). Additionally, minor amounts of limonene (6.38%) and β-myrcene (3.89%) were observed, adding new insights to the product profile of this enzyme (Fig. 3, Table 3). The TPS CsTPS19BL, when incubated with GPP, produced a mixture of monoterpenes, with the major products being (+)-linalool (54.12%) and (-)-linalool (40.87%) (Fig. 3, Table 3). These results align with the previous studies that have suggested this enzyme's strong preference for linalool production when GPP is present in the reaction (*Zager et al., 2019*). The TPS CsTPS20CT was previously characterised as a sesquiterpene synthase, with its primary product being hedycaryol when incubated with FPP (*Zager et al., 2019*). However, when incubated with GPP in this study, CsTPS20CT displayed an unexpected monoterpene production profile, forming geraniol as the dominant product (87.64%) (Fig. 3, Table 3).

### Sesquiterpene synthases (FPP assay)

All previously proposed sesquiTPSs analysed in this study were found to be active with FPP. CsTPS9FN produced a diverse range of sesquiterpenes, with epi-β-caryophyllene being the predominant product (67.30%) with β-caryophyllene (2.76%), humulene

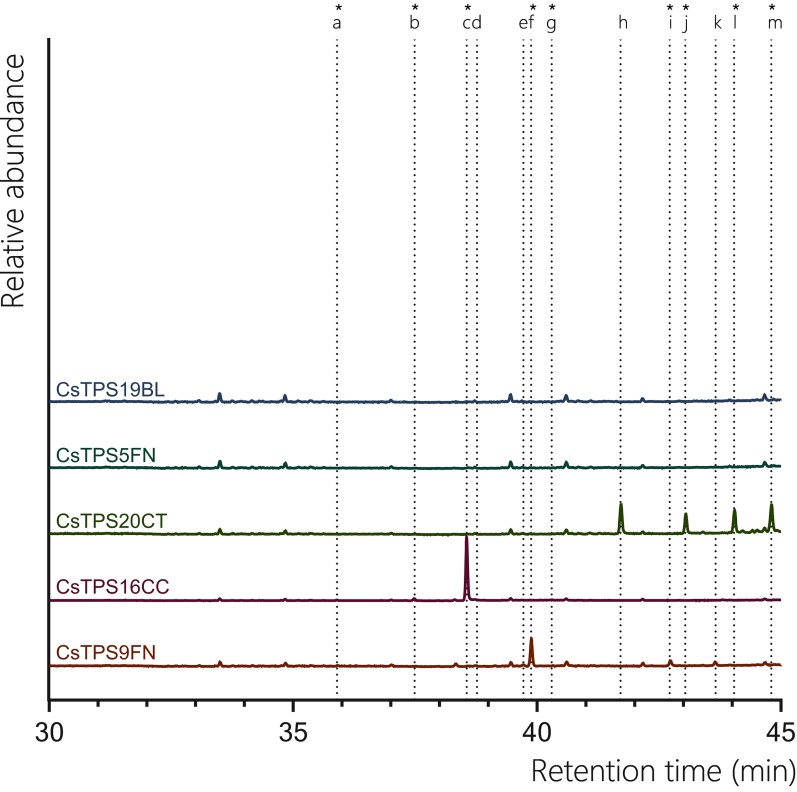

**Figure 4 GC-MS traces showing the sesquiterpene products of CsTPS.** Traces show GC-MS total ion chromatogram from CsTPS assays with FPP. CsTPS9FN: β-caryophyllene/humulene synthase; CsTPS16CC: Germacrene B synthase; CsTPS20CT: Hedycaryol synthase; CsTPS5FN: Myrcene/pinene synthase; CsTPS19BL: Nerolidol/Linalool synthase. Peaks: (a) β-elemene*, (b) γ-elemene*, (c) germacrene B*, (d) β-caryophyllene, (e) humulene, (f) epi-β-caryophyllene*, (g) alloaromadendrene*, (h) nerolidol* (i) elemol*, (j) germacrene-D*, (k) guaiol, (l) globulol*, (m) γ-eudesmol*, (n) α-eudesmol*. *No reference standard available, putative identification of compound using National Institute of Standards and Technology (NIST) library.

(4.49%) found to be present in low abundance (Fig. 4, Table 3). Additionally, germacrene D (15.26%) and globulol (10.19%) were also detected, both of which were not previously reported for this enzyme (Fig. 4, Table 3; *Booth, Page & Bohlmann, 2017*). This contrasts previous studies where β-caryophyllene and α-humulene were the only products reported. Consistent with previous studies, when incubated with FPP, CsTPS20CT primarily produced hedycaryol (which was detected as elemol due to GC-MS thermal degradation) (31.42%) (Fig. 4, Table 3; (*Zager et al., 2019*)). Other significant sesquiterpene products included γ-eudesmol (20.51%), α-eudesmol (28.11%), and guaiol (19.96%). This broad range of products indicates that CsTPS20CT is a multi-product sesquiterpene synthase, capable of producing a diverse array of eudesmol-type sesquiterpenes. The TPS CsTPS16CC produced γ-elemene (92.73%) as the dominant sesquiterpene when incubated with FPP, consistent with the enzyme's sesquiterpene synthase activity (Fig. 3, Table 3). Other minor products included β-elemene (3.82%) and germacrene B (1.05%), further supporting its activity toward forming elemene-type sesquiterpenes. Interestingly,

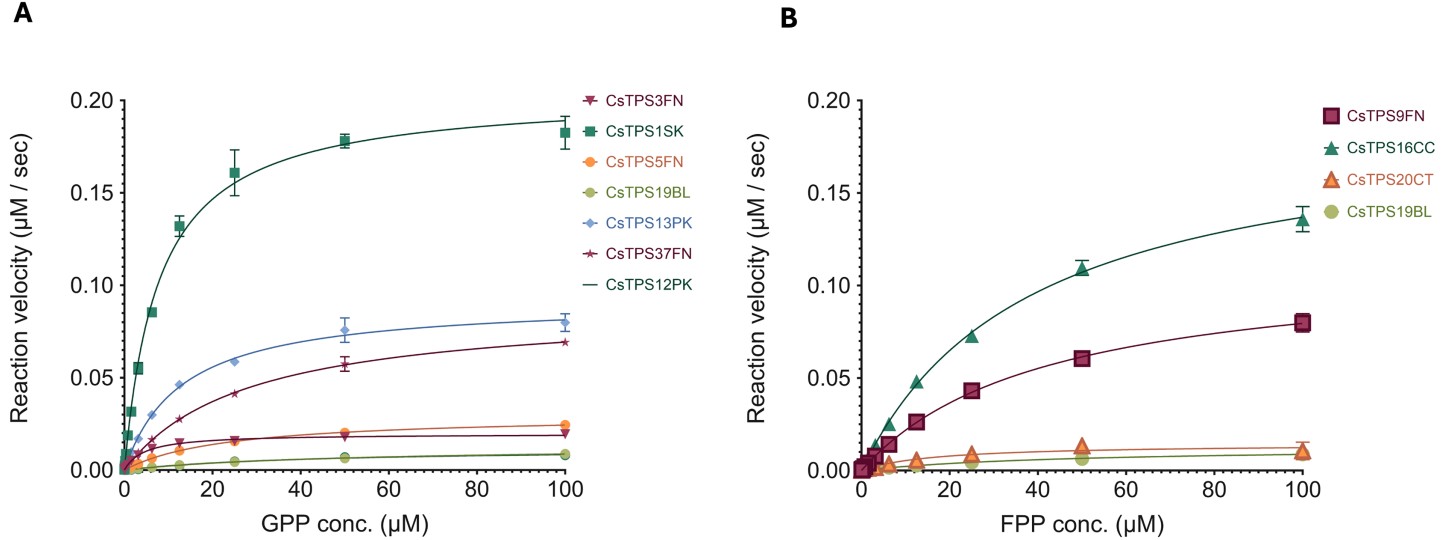

**Figure 5 Michaelis-Menten kinetics of *Cannabis sativa* terpene synthases (CsTPS).** (A) Non-linear regression analysis of steady-state kinetic assays for CsTPS enzymes using geranyl diphosphate (GPP) as the substrate, showing the rate of GPP catalysis (μM of GPP consumed per second). CsTPS3FN: β-myrcene synthase; CsTPS16CC: Germacrene B synthase; CsTPS1SK: Limonene synthase; CsTPS5FN: Myrcene/pinene synthase; CsTPS19BL: Nerolidol/Linalool synthase; CsTPS13PK: Ocimene synthase; CsTPS37FN: Terpinolene synthase; CsTPS12PK: Terpinene synthase. (B) Non-linear regression analysis of steady-state kinetic assays for CsTPS enzymes using farnesyl diphosphate (FPP) as the substrate, showing the rate of FPP catalysis (μM of FPP consumed per second). Each curve represents the data fitted to the Michaelis-Menten equation to determine kinetic parameters. CsTPS9FN: β- caryophyllene/humulene synthase; CsTPS16CC: Germacrene B synthase; CsTPS20CT: Hedycaryol synthase; CsTPS5FN: Myrcene/pinene synthase; CsTPS19BL: Nerolidol/Linalool synthase.

alloaromadendrene (1.06%) was also detected, which had not been previously associated with this enzyme. CsTPS5FN, when incubated with FPP, resulted in the exclusive production of farnesol (100%), a linear sesquiterpene alcohol (Fig. 4, Table 3). Farnesol had previously been reported as the single sesquiterpene product of this enzyme, highlighting the dual activity of CsTPS5FN with GPP and FPP as substrates (*Booth, Page & Bohlmann, 2017*). This enzyme's ability to produce both monoterpenes and sesquiterpenes makes it a versatile target for biotechnological applications. Finally, the TPS CsTPS19BL produced 100% nerolidol when incubated with FPP, consistent with prior studies (Fig. 4, Table 3; (*Zager et al., 2019*)). These findings demonstrate that each TPS exhibits a distinct product profile, with some producing a diverse array of sesquiterpenes and others generating a single, consistent product, highlighting the enzyme-specific nature of terpene biosynthesis.

## Kinetic properties of CsTPS enzymes

To better inform protein engineering strategies for TPSs, there is a need to understand their catalytic activity. We chose a high-throughput enzyme assay that can be performed without specialised equipment which is based on phosphate being produced. The kinetic profiles of the CsTPS are shown in Fig. 5. Substrate concentrations (FPP, GPP) ranged from 0 to 100 μM. The calculated $k_{cat\ ap}$ for each CsTPS was found to be between 0.0011 and 0.0204 s$^{-1}$ (Table 4).

**Table 4 Steady-state kinetic parameters for selected *Cannabis sativa* terpene synthases (CsTPS).**

| TPS Enzyme | Substrate | $K_m$ (µM) | $V_{max}$ (µM$^{-1}$ s$^{-1}$) | $k_{cat}$ (s$^{-1}$) |
|---|---|---|---|---|
| CsTPS3FN | GPP | 4.569 ± 0.411 | 0.0196 ± 0.0005 | 0.0020 |
| CsTPS9FN | FPP | 41.7 ± 3.73 | 0.1127 ± 0.0047 | 0.0113 |
| CsTPS16CC | FPP | 38.43 ± 2.83 | 0.1895 ± 0.0063 | 0.0190 |
| CsTPS20CT | FPP | 16.86 ± 6.42 | 0.0144 ± 0.0022 | 0.0014 |
| CsTPS1SK | GPP | 7.809 ± 0.678 | 0.2038 ± 0.0053 | 0.0204 |
| CsTPS5FN | GPP | 23.3 ± 1.34 | 0.0300 ± 0.0007 | 0.0030 |
| CsTPS19BL | GPP | 48.45 ± 4.39 | 0.0129 ± 0.0006 | 0.0013 |
| | FPP | 17.32 ± 5.23 | 0.0102 ± 0.0002 | 0.0011 |
| CsTPS12PK | GPP | 41.85 ± 8.29 | 0.0119 ± 0.0011 | 0.0012 |
| CsTPS13PK | GPP | 12.96 ± 1.23 | 0.0918 ± 0.0029 | 0.0092 |
| CsTPS37FN | GPP | 27.71 ± 1.92 | 0.0884 ± 0.0025 | 0.0088 |

The kinetic analysis revealed a broad range of affinities and turnover rates across the different CsTPS variants, reflecting their diverse roles in terpene biosynthesis. For instance, CsTPS1SK exhibited a relatively low $K_m$ for GPP (7.809 ± 0.678 µM) paired with a high $V_{max}$ (0.2038 ± 0.0053 µM$^{-1}$ s$^{-1}$) and $k_{cat}$ (0.0204 s$^{-1}$), indicating a strong substrate affinity and rapid catalysis, which suggests its efficiency in monoterpene production. On the other hand, CsTPS13PK which also utilizes GPP, displayed a moderately higher $K_m$ (12.96 ± 1.23 µM) with a lower $V_{max}$ (0.0918 ± 0.0029 µM$^{-1}$ s$^{-1}$), signifying a slightly reduced affinity for GPP compared to CsTPS1SK but still an efficient monoterpene producer.

For enzymes using FPP as a substrate, CsTPS9FN exhibited the highest $K_m$ (41.7 ± 3.73 µM), indicating a lower affinity for FPP compared to CsTPS20CT, which had a significantly lower $K_m$ (16.86 ± 6.42 µM). However, CsTPS9FN also showed a much higher $V_{max}$ (0.1127 ± 0.0047 µM$^{-1}$ s$^{-1}$), than CsTPS20CT (0.0144 ± 0.0022 µM$^{-1}$ s$^{-1}$), suggesting distinct catalytic efficiencies and roles in sesquiterpene biosynthesis. This variability highlights the challenge of predicting enzyme function based solely on sequence similarity and the need for functional and kinetic characterisation. By elucidating these kinetic properties, our study provides a foundation to understand which CsTPS enzymes efficiently produce specific terpenes. This can be used to inform engineering of this class of enzymes for targeted terpene production.

## Development of a targeted TPS crystallisation screen

Currently, many macromolecular crystallisation screens are commercially available; however, relatively few are specifically designed for particular protein families. We created a 48-condition screen, called the TPS screen, with conditions specifically directed towards the crystallisation of the TPS protein family by analysing the crystallisation conditions for all TPS and TPS-related proteins identified in the PDB. Most TPS crystals were obtained between pH 6.0 and 6.9 (34.4%), with the next highest range from pH 7.0 to 7.9 (23.4%). The majority of TPS crystallised in Bis-Tris buffer (Figs. 6A, 6B), with 24.6% of conditions ranging widely in pH from 3.8 to 10 (Figs. 6A, 6B). Tris was the next most common buffer used to obtain TPS crystals (19.7%). Combining this data with the pH data resulted in the

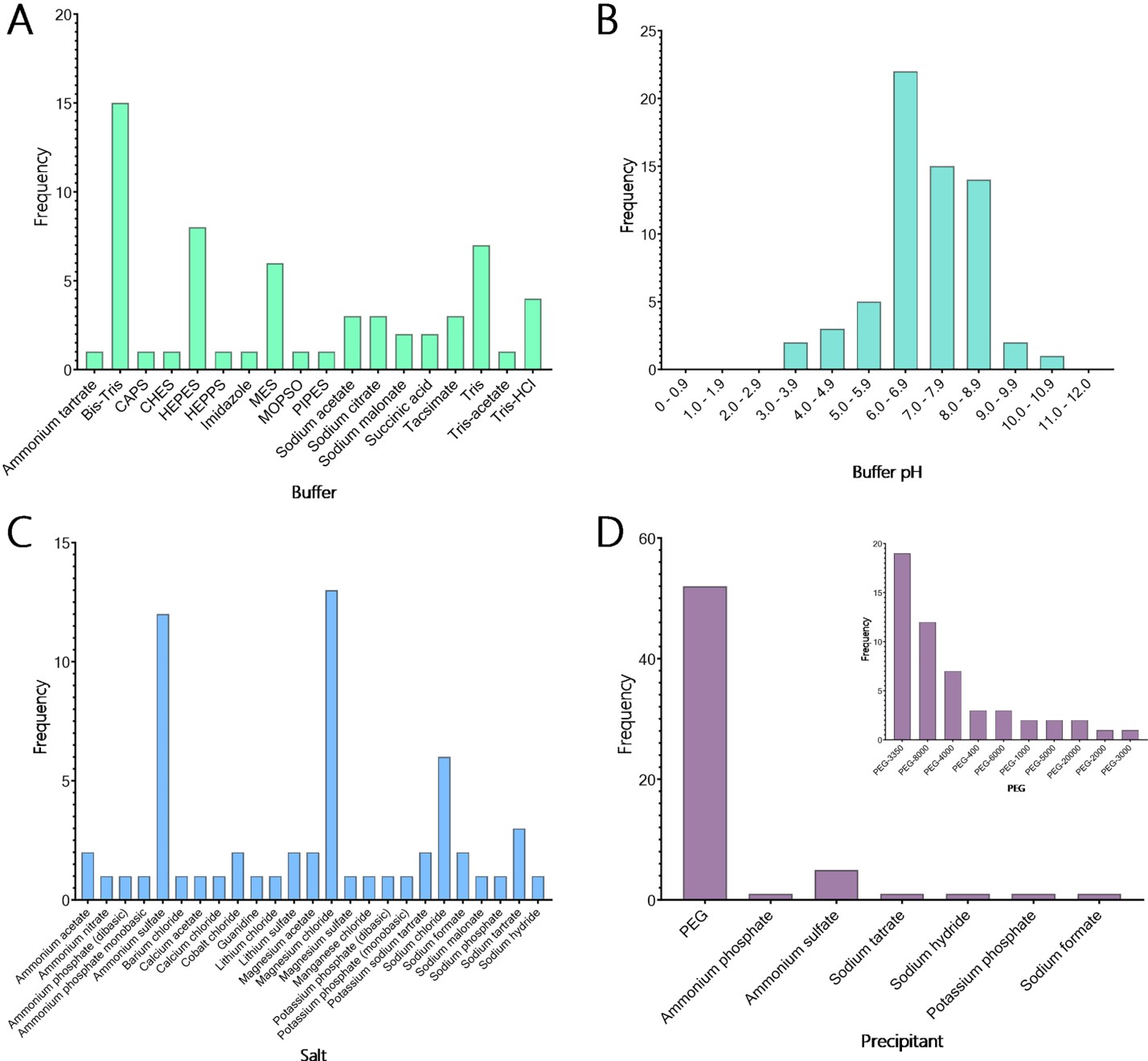

**Figure 6 Crystallisation conditions for terpene synthase proteins.** Data mining of the Protein Data Bank (PDB; *Berman et al., 2000*) yielded conditions used in the crystallisation of many terpene synthases. (A) The buffers used in 104 crystallisation conditions shows that the majority of terpene synthases crystallise with Bis-Tris. The next most common buffers were Tris, HEPES and MES. (B) The predominant pH used shows that the majority of terpene synthases crystallise between pH 6.0 and 6.9, with an overall preference for slightly basic pH. (C) In the conditions that reported salts a total of 26 different salts were used, with most conditions containing magnesium chloride, ammonium sulphate and sodium chloride in the crystallisation of terpene synthase proteins. (D) Of the crystallisation conditions that reported the precipitant composition, 84% contained some variation of PEG (inset). Of these conditions, there is a strong preference for PEG-3350 and PEG-8000.

selection of six buffer/pH combinations for use in the TPS-directed screen. These buffers are Bis-Tris (pH 6.0), Bis-Tris (pH 6.5), Bis-Tris (pH 7.0), Tris (pH 7.5), Tris (pH 8.0) and Tris (pH 8.5). Of the 104 crystallisation conditions identified for TPS crystal production, 50 conditions reported the inclusion of salts to crystallise the molecule (Fig. 6C). Although 22 different salts were used in these crystallisation conditions, a large majority (26%) of these conditions contained $MgCl_2$. This is consistent with TPS functioning as metalloenzymes, where $Mg^{2+}$ or $Mn^{2+}$ plays a crucial role in stabilising the $PP_i$ substrates leaving group and facilitating catalysis. NaCl was the next most common salt used and was seen in 12% of crystallisation conditions. This data led to the selection of both magnesium chloride and sodium chloride as counter ions in the TPS-directed screen.

The final components of the analysed crystallisation conditions were precipitants, which were provided for 62 of the 104 conditions. Of the 16 different precipitants used, various molecular weights of polyethylene glycol (PEG) comprised 83.9% of the conditions (Fig. 6D (Inset)). The next most used precipitant was ammonium sulphate, which was used in 8% of the conditions, indicating a strong preference for PEG in the crystallisation of TPS proteins. When examining the 52 conditions containing PEG in more depth (Fig. 6D), PEG-3350 and PEG-8000 were most often used (30.6% and 19.4% of conditions, respectively). As a result, PEG-3350 and PEG-8000 were chosen to be included in the TPS-directed screen.

To create the final TPS screen, four concentrations (5, 15, 25 and 35%) of each molecular weight PEG (3,350 and 8,000) were chosen, along with 100 mM of each of the buffers (Bis-Tris pH 6.0, Bis-Tris pH 6.5, Bis-Tris pH 7.0, Tris pH 7.5, Tris pH 8.0 and Tris pH 8.5). Each condition additionally contains 200 mM $MgCl_2$ and 200 mM NaCl (Table S2). This systematic approach to screen design provides a balanced matrix of variables aimed at identifying conditions for TPS crystal formation. By incorporating a diverse yet targeted range of parameters, this screen facilitates the formation of crystals, addressing the challenges of obtaining high-quality crystals essential for structural characterisation and subsequent protein engineering efforts.

## Crystallisation and optimisation of CsTPS crystals

Each of the 10 CsTPS proteins were screened against the 48 conditions of the TPS-screen, as well as several other commercially available screens, all at room temperature. Initial crystal formation was observed as early as 24 h post-screen setup, in many conditions of the TPS-screen. All 10 proteins demonstrated varying degrees of crystallisation, with at least some form of precipitate forming in every case. Well-defined crystals were observed for CsTPS37FN, CsTPS1Sk, CsTPS3FN, CsTPS13PK, and CsTP12PK, while several of the other proteins produced spherical aggregates. Interestingly, with the exception of CsTPS1SK, all CsTPS formed crystals with either star or cubic morphologies. CsTPS1SK, on the other hand, consistently produced rod-shaped crystals (Fig. 7). The best crystals in terms of size and morphology were found in condition 16 of the TPS-screen (0.1 M Bis-Tris (pH 7.0), 25% PEG-3350, 0.2 M NaCl, 0.2 M $MgCl_2$), using a 2:1 ratio of CsTPS protein (10 mg ml$^{-1}$) to reservoir solution.

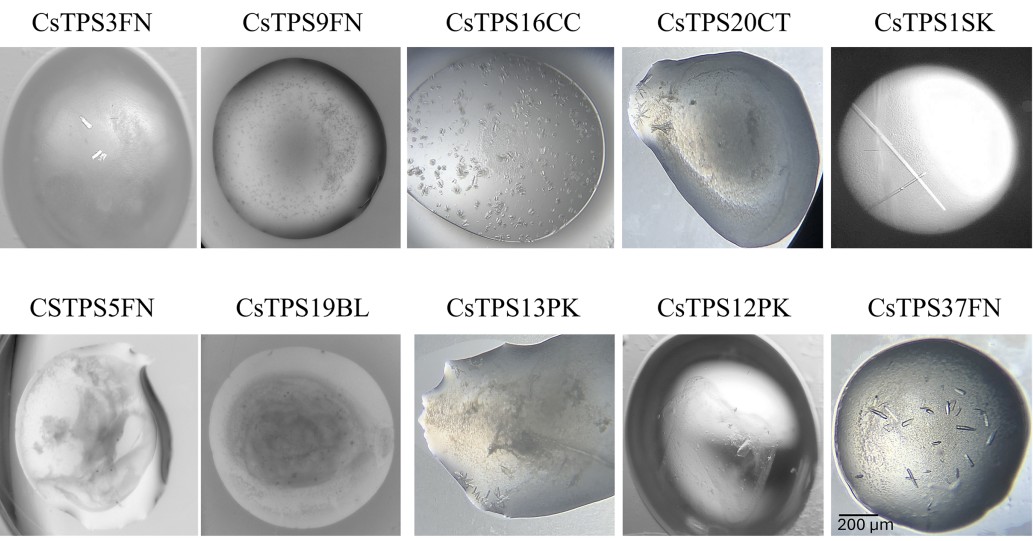

**Figure 7 Images of the best crystallisation conditions for recombinant terpene synthases from *Cannabis sativa* (CsTPS).** The recombinant terpene synthases were purified from *E. coli* and crystals were produced using the in-house targeted terpene synthase screen.

## DISCUSSION

The resin of *Cannabis sativa* is rich in mono- and sesquiterpenes, which are believed to contribute to its pharmacological effects. While much of the research on cannabis terpenes has focused on phytochemical composition for forensics and breeding, the molecular biology of terpene formation in cannabis has received less attention. Understanding the enzyme functions and active site architecture of terpene biosynthesis in *C. sativa* is key for genetic improvements aimed at optimising terpene profiles. The functional diversity of TPSs, driven by subtle amino acid changes in the active site, plays critical roles in plant defence and scent profiles and holds significant biotechnological potential for industries such as cosmetics, food, agrochemicals, and pharmaceuticals. Here we describe a systematic pipeline that begins with the recombinant over-expression and characterisation of *C. sativa* TPS proteins and aims at facilitating targeted terpene production. Existing protocols for overexpressing TPSs in bacterial systems often face challenges such as protein misfolding, low yield, and instability. Our pipeline addresses these challenges by optimising factors like protein solubility, stability, and activity. This work builds on previous work by *Allen et al. (2019)*, *Booth et al. (2020)*, *Booth, Page & Bohlmann (2017)*, *Booth & Bohlmann (2019)*, *Günnewich et al. (2007)*, *Xu et al. (2024)*, *Zager et al. (2019)*, which identified at least 55 distinct TPS genes in the *C. sativa* genome, functionally validating 38 of them (*Wiles et al., 2022*). Key insights, such as optimising buffer conditions to prevent protein aggregation, address the stability issues often encountered in the expression of plant-derived enzymes. Although this expression system offers a promising pathway for scaling up terpene production, further optimisation may be necessary to improve yields and stability for specific enzymes, as seen in similar studies of other plant-derived enzymes (*Belwal, Georgiev & Al-Khayri, 2022*; *Moon et al., 2020*).

Understanding the molecular factors that govern enzyme expression in heterologous systems will be crucial for maximising production yields and integrating these enzymes into biotechnological workflows.

The functional characterisation of CsTPS enzymes presented in this work offers valuable insights into the catalytic properties, substrate specificities and product variability of distinct CsTPSs. Our findings demonstrate the considerable diversity in product profiles of CsTPS enzymes, as exemplified by limonene synthase (CsTPS1SK) and terpinolene synthase (CsTPS37FN) both of which utilise a common substrate (GPP) to produce a distinct dominant terpene product (limonene or terpinolene, respectively) alongside a myriad of other shared and distinct monoterpene products. Interestingly, CsTPS3FN exhibited strict specificity for β-myrcene, producing this monoterpene at 100% when incubated with GPP, while CsTPS19BL generated a mixture of linalool isomers, but exclusively produced nerolidol when incubated with FPP, highlighting these enzymes as candidates for pure, single-terpene synthesis in microbial factories and for investigating active site residues that drive single-product formation, informing future protein engineering applications (*Booth, Page & Bohlmann, 2017*; *Zager et al., 2019*). In contrast, CsTPS19BL exhibited broader versatility, synthesising both the monoterpene alcohol linalool and the sesquiterpene alcohol nerolidol by catalysing two different substrates (GPP and FPP, respectively). This versatility highlights the complexity of terpene biosynthesis, a theme also observed in other studies (*Booth, Page & Bohlmann, 2017*), where closely related TPS enzymes exhibit distinct functional behaviours due to subtle differences in active site residues.

Importantly, our results reinforce that genetic sequence alone is insufficient to predict TPS function, necessitating biochemical validation. Furthermore, despite high sequence homology, CsTPS homologs from different plant species display distinct product profiles, as exemplified by CsTPS1SK, which, despite its similarity to limonene synthases from *Abies grandis* and *Mentha spicata* (*Bohlmann, Steele & Croteau, 1997*; *Hyatt et al., 2007*; *Srividya, Lange & Lange, 2020*), produces a different terpene spectrum. This divergence likely reflects evolutionary adaptations to ecological niches, emphasising the evolutionary plasticity of TPS enzymes. Beyond genetic variation, post-translational modifications and assay conditions may further modulate catalytic activity, as demonstrated by our characterisation of β-caryophyllene/α-humulene synthase (CsTPS9FN), which exhibited a broader product range than previously reported. This finding aligns with prior studies (*Zager et al., 2019*) who also reported condition-dependent differences in CsTPS enzyme activity, particularly CsTPS19BL, responsible for producing linalool and nerolidol. Our results confirmed nerolidol production when incubated with FPP and linalool with GPP, consistent with (*Zager et al., 2019*) who documented condition-dependent shifts in CsTPS enzyme activity, particularly CsTPS19BL, whose linalool and nerolidol production varied with assay conditions. Our results confirm that CsTPS19BL produces nerolidol from FPP and linalool from GPP, yet differences in product ratios suggest an interplay between genetic background and experimental parameters.

The catalytic turnover rates (*kcat*) of CsTPS enzymes, ranging from 0.0011 to 0.0204 s$^{-1}$, are consistent with the generally slow kinetics of TPS enzymes (*Günnewich et al., 2007*).

Nevertheless, the observed variations in substrate affinity (Km) among CsTPS isoforms suggest functional specialisation tailored to distinct physiological roles. When compared to thermostable TPSs that exhibit higher catalytic efficiencies (*Styles et al., 2017*), CsTPS enzymes appear prioritise substrate specificity and structural flexibility over extreme stability or turnover, reflecting adaptation to mesophilic plant environments.

Structurally TPS enzymes exhibit a large amount of conservation with minor differences in their active sites resulting in their broad catalytic and functional repertoires. Comparisons to other structurally characterised TPSs, such as *Artemisia annua* β-farnesene synthase (AaFS) and *Nicotiana tabacum* 5-epi-aristolochene synthase (TEAS), also analysed using the malachite green assay (*Vardakou et al., 2014*), suggest that even minor alterations in TPS active sites can lead to substantial changes in *kcat* and *Km* values. These insights reinforce the necessity of optimising expression systems and assay conditions for precise TPS characterisation. Overall, this study provides essential insights into the factors that influence terpene production in *C. sativa* that will inform the optimisation of these enzymes for large-scale production of specialised metabolites.

The development of the TPS-crystallisation screen represents a significant advancement in the structural characterisation of CsTPS enzymes, that will facilitate understanding of their catalytic mechanisms and enabling the future design of engineered variants with enhanced activity. This work builds on previous structural studies of plant TPS enzymes, such as limonene synthases from *Mentha spicata* and *Citrus sinensis* (*Hyatt et al., 2007*; *Kumar et al., 2017*; *Morehouse et al., 2017*), 1,8-cineole synthase from *Salvia fruticose* (*Kampranis et al., 2007*), taxadiene synthase from *Taxus brevifolia* (*Köksal et al., 2011*), abietadiene synthase from *Abies grandis* (*Zhou et al., 2012*), α-bisabolol synthase from *Artemisia annua* (*Li et al., 2013*), and 5-epi-aristolochene synthase from *Nicotiana tabacum* (*Starks et al., 1997*), that allowed us to optimise crystallisation conditions for CsTPS, yielding crystals and high-resolution diffraction data. This systematic approach, akin to that of *Pryor, Wozniak & Hollis (2012)* highlights the adaptability of developing a narrow range of crystallisation conditions for specific protein families. The crystallisation pipeline developed here not only aids in the study of TPSs but also sets a precedent for the structural analysis of other enzyme families.

## CONCLUSIONS

The variety and versatility of TPS enzymes provides a significant challenge to their study, whilst also providing valuable opportunities for the production of industrially-important terpenes. The pipeline presented in this study addresses these key challenges by optimising TPS solubility, stability, and enzymatic activity, establishing a framework for scalable, specific terpene production. Beyond advancing the TPS field, this approach has broader applications in plant secondary metabolism research and the engineering of plant enzyme families with industrial potential.

## ACKNOWLEDGEMENTS

We would like to acknowledge Myrna Deseo for her assistance with the gas-chromatography mass spectrometry, and the La Trobe Proteomic and Metabolomic

Platform for their assistance in gel band protein mass spectrometry (LC-MS/MS) and the provision of instrumentation and technical support.

### Funding
This research was supported by the AustralianResearch Council (ARC) Research Industrial Transformation ResearchHub for Medicinal Agriculture (IH180100006). The funders had no role in study design, data collection and analysis, decision to publish, or preparation of the manuscript.

### Grant Disclosures
The following grant information was disclosed by the authors:
AustralianResearch Council (ARC).
Research Industrial Transformation ResearchHub for Medicinal Agriculture: IH180100006.

### Competing Interests
Travis Beddoe is an Academic Editor for PeerJ.

### Author Contributions

- Danielle Wiles conceived and designed the experiments, performed the experiments, analyzed the data, prepared figures and/or tables, authored or reviewed drafts of the article, and approved the final draft.
- James Roest performed the experiments, analyzed the data, authored or reviewed drafts of the article, and approved the final draft.
- Bhuvana Shanbhag conceived and designed the experiments, analyzed the data, authored or reviewed drafts of the article, and approved the final draft.
- Julian Vivian performed the experiments, analyzed the data, authored or reviewed drafts of the article, and approved the final draft.
- Travis Beddoe conceived and designed the experiments, analyzed the data, authored or reviewed drafts of the article, and approved the final draft.

### Data Availability
The raw data is available in the Supplemental File.

### Supplemental Information
Supplemental information for this article can be found online at http://dx.doi.org/10.7717/peerj.19723#supplemental-information.

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
