# Peer review of "Integrated platform for structural and functional analysis of terpene synthases of Cannabis sativa"

_PeerJ, doi:10.7717/peerj.19723_

## Round 0.1 · original submission · Major Revisions

Please consider all the reviewer's suggestions and prepare a revised version along with a rebuttal letter.

·

Basic reporting

The writing and use of English are clear and unambiguous. The cited literature is appropriate for the field of study concerning TPS enzymes. Tables, figures, and supplementary materials are available, and the raw data are consistent with those presented in the manuscript. However, some figures should be edited for ease of understanding as Figure S2:
*Please include the axis labels for the graphs shown in Figure S2. For example, does the X-axis correspond to time or retention volume?
*Supplementary Table 1 listed in lines 281 and 287, must be Supplementary Table S2

Experimental design

In general, all the methods are clearly described and provide the necessary information for comparison and, if necessary, replication. I only have one comment in the section on Crystallization and Optimization of TPS crystals:
* Line 286, Please indicate the exact temperature for incubating the crystallization plates.

Validity of the findings

* Lines 482-486..The optimal crystallization condition selected based on the CsTPS crystal size and shape parameters was condition 16. You mention that this condition was optimized, considering several variables (precipitating agent, etc.). Please show the results of the crystallization optimization.

*Lines. 548-560...The discussion regarding the structural adaptation and comparison of CsTPS with other TPSs needs more information related to the structural component of this type of enzymes. I suggest you improve the description in this section to justify the discussion regarding the possible relationship between the Michaelis-Menten constants and the structural adaptations of CsTPS.

* Lines 562-576...The selection criteria for the best CsTPS crystallization conditions were based on crystal size and shape. However, I suggest that these crystals should be subjected to X-ray diffraction experiments to gain more information about their quality (diffraction pattern) and if they are suitable for TPS structural analysis.

Additional comments

no comment

Reviewer 2 ·

Basic reporting

no comment

Experimental design

Why were different cultivars of C. sativa considered?

Validity of the findings

no comment

Additional comments

no comment

Annotated reviews are not available for download in order to protect the identity of reviewers who chose to remain anonymous.

---

## Round 0.2 · accepted · Accept

Thanks for addressing all the reviewers' comments.

·

Basic reporting

No comment.

Experimental design

No comment.

Validity of the findings

No comment.

Additional comments

No comment.

Reviewer 2 ·

Basic reporting

no comment

Experimental design

no comment

Validity of the findings

no comment

Additional comments

The authors have improved the manuscript. I don't have any further comments, and I recommend to accept the manuscript for publication.